# Heterogeneous nuclear ribonucleoprotein E1 binds polycytosine DNA and monitors genome integrity

Bidyut K Mohanty[1], Joseph AQ Karam[1], Breege V Howley[1], Annamarie C Dalton[1], Simon Grelet[1,2], Toros Dincman[3], William S Streitfeld[1], Je-Hyun Yoon[1], Lata Balakrishnan[4], Walter J Chazin[5], David T Long[1], Philip H Howe[1,2]

Heterogeneous nuclear ribonucleoprotein E1 (hnRNP E1) is a tumor suppressor protein that binds site- and structure-specifically to RNA sequences to regulate mRNA stability, facilitate alternative splicing, and suppress protein translation on several metastasis-associated mRNAs. Here, we show that hnRNP E1 binds polycytosine-rich DNA tracts present throughout the genome, including those at promoters of several oncogenes and telomeres and monitors genome integrity. It binds DNA in a site- and structure-specific manner. hnRNP E1-knockdown cells displayed increased DNA damage signals including γ-H2AX at its binding sites and also showed increased mutations. UV and hydroxyurea treatment of hnRNP E1-knockdown cells exacerbated the basal DNA damage signals with increased cell cycle arrest, activation of checkpoint proteins, and monoubiquitination of proliferating cell nuclear antigen despite no changes in deubiquitinating enzymes. DNA damage caused by genotoxin treatment localized to hnRNP E1 binding sites. Our work suggests that hnRNP E1 facilitates functions of DNA integrity proteins at polycytosine tracts and monitors DNA integrity at these sites.

## Introduction

Genome instability is a hallmark of cancer (Negrini et al, 2010). Cells are constantly exposed to various exogenous agents such as UV, X-rays, and chemicals, and endogenous agents such as reactive oxygen species that can damage DNA and cause genome instability (Friedberg, 2008; Chatterjee & Walker, 2017). DNA secondary structures such as G-quadruplexes (G4s) formed by polyguanine (poly-G) tracts also play important regulatory roles in DNA transactions and genome integrity (Bochman et al, 2012; Saini et al, 2013; Vasquez & Wang, 2013; Varshney et al, 2020). Poly-G/poly-C sequences are present at promoter proximal regions of several oncogenes including *c-MYC*, *BCL2*, *EGFR*, *HIF1-α*, *PDGF*, and *VEGF* and at telomeres (Siddiqui-Jain et al, 2002; Dai et al, 2006a, 2006b; Qin et al,

2007; Sun et al, 2011; Greco et al, 2017). Various cellular processes that involve breaks in DNA or DNA-free ends, including replication, repair, recombination, transcription, and related cell cycle progression, have the potential to cause genome instability (Aguilera & García-Muse, 2013; Tubbs & Nussenzweig, 2017).

Cells have developed sophisticated mechanisms such as DNA damage response (DDR) to monitor and repair DNA damage (Zhou & Elledge, 2000). Upon DNA damage or replication blockage, a battery of checkpoint proteins including sensors, adaptors, and effectors are activated and halt cell cycle progression (Harrison & Haber, 2006). Various DNA repair processes operate in the cell (Friedberg, 2008; Choi et al, 2015). A key intermediate of DNA damage/repair and replication processes is the generation of single-stranded DNA (ssDNA), which invites the heterotrimeric protein namely replication protein A (RPA); RPA coats ssDNA to protect it and in addition, it leads to several processes as descried below (Wold, 1997; Maréchal & Zou, 2015; Sugitani & Chazin, 2015; Caldwell & Spies, 2020). Checkpoint proteins ATR-ATRIP are recruited at damage sites by RPA-coated ssDNA (Choi et al, 2010). RPA colocalizes with γ-H2AX at IR- and HU-induced double strand breaks in DNA (Balajee & Geard, 2004). During DNA replication millions of Okazaki fragments are synthesized in the lagging strand (Balakrishnan & Bambara, 2013). Okazaki fragment maturation involves removal of single-stranded RNA-DNA flap by Fen1 endonuclease and RNase HI, which is regulated by RPA (Bae et al, 2001; Chai et al, 2003; Zaher et al, 2018). Upon DNA damage by UV and methyl methanesulfonate (MMS), the DNA clamp proliferating cell nuclear antigen (PCNA) gets monoubiquitinated and loads mutagenic or nonmutagenic DNA polymerases at DNA repair sites; PCNA monoubiquitination requires RPA (Niimi et al, 2008). During nucleotide excision repair, interaction between RPA and XPA orients the latter on DNA (Topolska-Woś et al, 2020). Hydroxyurea (HU) treatment reduces the nucleotide pool in the cell, which uncouples replicative helicase and DNA polymerase thereby generating stretches of ssDNA; ssDNA binding proteins such as RPA play important role in protecting the ssDNA (Balajee & Geard, 2004; Alvino et al, 2007; Papadopoulou et al, 2015; Singh & Xu, 2016).

[1]Department of Biochemistry and Molecular Biology, Medical University of South Carolina, Charleston, SC, USA   [2]Hollings Cancer Center, Medical University of South Carolina, Charleston, SC, USA   [3]Division of Hematology and Oncology, Department of Medicine, Medical University of South Carolina, Charleston, SC, USA   [4]Department of Biology, School of Science, Indiana University Purdue University Indianapolis, Indianapolis, IN, USA   [5]Departments of Biochemistry and Chemistry and Center for Structural Biology, Vanderbilt University, Nashville, TN, USA

Correspondence: howep@musc.edu

All these findings underscore the importance of RPA and other ssDNA binding proteins in DNA integrity.

Heterogeneous nuclear ribonucleoprotein E1 (hnRNP E1, PCBP1, or αCP1) has been studied extensively for its RNA binding and transactions on RNAs (Chaudhury et al, 2010a; Grelet & Howe, 2019). A 37-kD protein with 356 amino acids, it contains three K-homology domains of ~70 amino acids, namely, KH1 (aa 13–86), KH2 (aa 97–169), and KH3 (aa 280–355) (Leffers et al, 1995). The protein binds to 3′-UTRs of several mRNAs in sequence- and structure-specific manner to regulate protein translation (Chaudhury et al, 2010b; Hussey et al, 2012). It also binds to a structural element located in exon 1 of PNUTS (also known as PPP1R10) pre-RNA to regulate alternative splicing (Grelet et al, 2017). The mechanism of RNA binding and translational suppression by hnRNP E1 is known (Chaudhury et al, 2010b; Hussey et al, 2011, 2012). It binds site-specific structural motifs (TGFβ-activated translation RNA; BAT RNA) present in 3′ UTRs of mRNAs to inhibit translation elongation on the metastasis-associated mRNAs (Chaudhury et al, 2010b; Hussey et al, 2011). High-throughput sequencing of hnRNP E1-bound RNA sequences led to identification of a consensus BAT element (Fig 1A) that binds to hnRNP E1 protein (Brown et al, 2015, 2016). The consensus BAT element contains three "rCrCrC" repeats, and a point mutation in any of the first "rC"s of the "rCrCrC" repeats compromised hnRNP E1 binding to the consensus BAT RNA (Brown et al, 2016), underscoring the importance of cytosine bases in its RNA binding.

hnRNP E1 protein is present in both the cytoplasm and nucleus of human cells (Chkheidze & Liebhaber, 2003; Fujimura et al, 2009; Ghanem et al, 2014), suggesting that the protein has nuclear function(s). We predicted that hnRNP E1 should bind to BAT RNA-like single-strand DNA (ssDNA) sequences and regulate DNA transactions. Although hnRNP E1 has been localized to telomeres, no function has been assigned to its DNA binding (Chen et al, 2012). Here, we show that hnRNP E1 binds to poly-C DNA tracts including those present at promoter proximal regions of various oncogenes and telomeres and localizes to promoter proximal regions of several genes and telomeres in the cell. In silico analysis shows that the genome contains numerous potential hnRNP E1 binding poly-C tracts. hnRNP E1 binds to secondary structures generated by poly-C tracts and plays an important role in the regulation of secondary structures such as i-motifs and G4s. Absence of hnRNP E1 activated basal DNA damage signals, which were exacerbated by HU and UV; DNA damage in hnRNP E1 knockdown (E1KD) cells localized to some of the hnRNP E1-binding sites tested. E1KD cells also displayed an increase in mutations. We propose that poly-C tract DNA binding is one of the mechanisms by which hnRNP E1 monitors genome integrity.

# Results

### hnRNP E1 binds to numerous poly-C–rich DNA tracts

hnRNP E1 binds polypyrimidine tracts in BAT RNA elements present at 3′ UTRs of several mRNAs and regulates protein translation (Chaudhury et al, 2010b). From an analysis of all the BAT RNA sequences, we had determined a consensus BAT RNA sequence that contained three "rCrCrC" tracts (Fig 1A) (Brown et al, 2016). Mutations of first "cytosine" bases in each tract of BAT RNA adversely affected hnRNP E1–RNA interaction (Brown et al, 2016) and this led us to predict that hnRNP E1 may also bind to ssDNA containing poly-C–rich tracts. hnRNP E1 protein has been found not only in the cytoplasm but also in the nucleus of human cells supporting a physiological role for hnRNP E1 in the nucleus (Chkheidze & Liebhaber, 2003; Fujimura et al, 2009; Ghanem et al, 2014). Our immunofluorescence analysis of A549 cells shows that hnRNP E1 is predominantly nuclear (Fig 1B). Moreover hnRNP E1 has been localized to telomeres suggesting a role for the protein in DNA transactions although no biological role has yet been assigned to hnRNP E1 on DNA (Chen et al, 2012). With all this information we wanted to analyze poly-C DNA binding by hnRNP E1. Because the promoter proximal regions of several oncogenes contain poly-G/poly-C tracts (Siddiqui-Jain et al, 2002; Dai et al, 2006a, 2006b; Qin et al, 2007; Sun et al, 2011; Greco et al, 2017), we predicted that hnRNP E1 would bind to these poly-C tracts. We examined chromosomal localization of hnRNP E1 in A549 cells by chromatin immunoprecipitation (ChIP) and observed that hnRNP E1 was specifically enriched at promoter-proximal regions of *c-MYC*, *h-RAS*, and *EGFR* (Fig 1C). We wanted to determine whether hnRNP E1 binds to poly-C strand, poly-G strand or their double-stranded DNA at these sequences. We conducted in vitro DNA binding assay. (electrophoretic mobility shift assay. - EMSA) with poly-C–rich tracts (Fig 1A) and their complementary poly-G tracts present at the promoter proximal regions of various oncogenes. As shown previously (Brown et al, 2016) purified GST-fusion protein of hnRNP E1 (Fig 1D) bound to the 43-base consensus BAT RNA (Fig 1E) in an RNA electrophoretic mobility shift assay (REMSA). In an EMSA assay, the protein bound to ssDNA containing poly-C tracts present in *c-MYC* promoter proximal region (Fig 1F), but not to its complementary poly-G strands (Fig 1G). Similarly, hnRNP E1 bound to the poly-C–rich strand of telomeres (TEL DNA C) (Fig 1H), but not to its complementary poly-G strand (Fig 1I). We examined binding of hnRNP E1 to several other poly-C–rich ssDNA sequences present at promoter proximal regions of *BCL2*, *HIF1-α*, *EGFR*, *PDGF*, and *VEGF* genes (Fig 1A). As shown (Fig 1J), hnRNP E1 bound specifically to all these poly-C–rich tracts albeit with different efficiencies. As will be shown later, the protein also bound to poly-C tracts present in the promoter proximal region of the h-RAS oncogene (Fig 2H). It may be noted that double-stranded DNA made from telomeric sequences did not bind to hnRNP E1 protein (data not shown). Quantification of protein:DNA and protein:BAT RNA interactions from the EMSA and REMSA assays show (Figs 1M, S1, and S2) that for a 50% shift of the DNA/RNA used, a three to fourfold molar excess protein over *c-MYC*, *PDGF*, *HIF1-α*, and *BCL2* (3.2, 3.6, 3.5, and 3.6-fold, respectively), a 4.4-fold excess protein over TEL DNA C, and a 11-fold excess protein over BAT RNA was needed. In contrast, up to a 48-fold excess of protein could not show a 50% shift of *VEGF* and *EGFR* DNA fragments; whereas 30-fold molar excess of the protein resulted in 25% shift of *VEGF*, 48-fold molar excess of the protein showed ~10% shift of *EGFR* poly-C DNA.

hnRNP E1 contains three KH domains, namely, KH1, KH2, and KH3, and we observed that all the three domains bound to BAT RNA albeit with different efficiency (Fig 1K). KH1 bound less efficiently to BAT RNA in comparison to the whole protein but it bound more

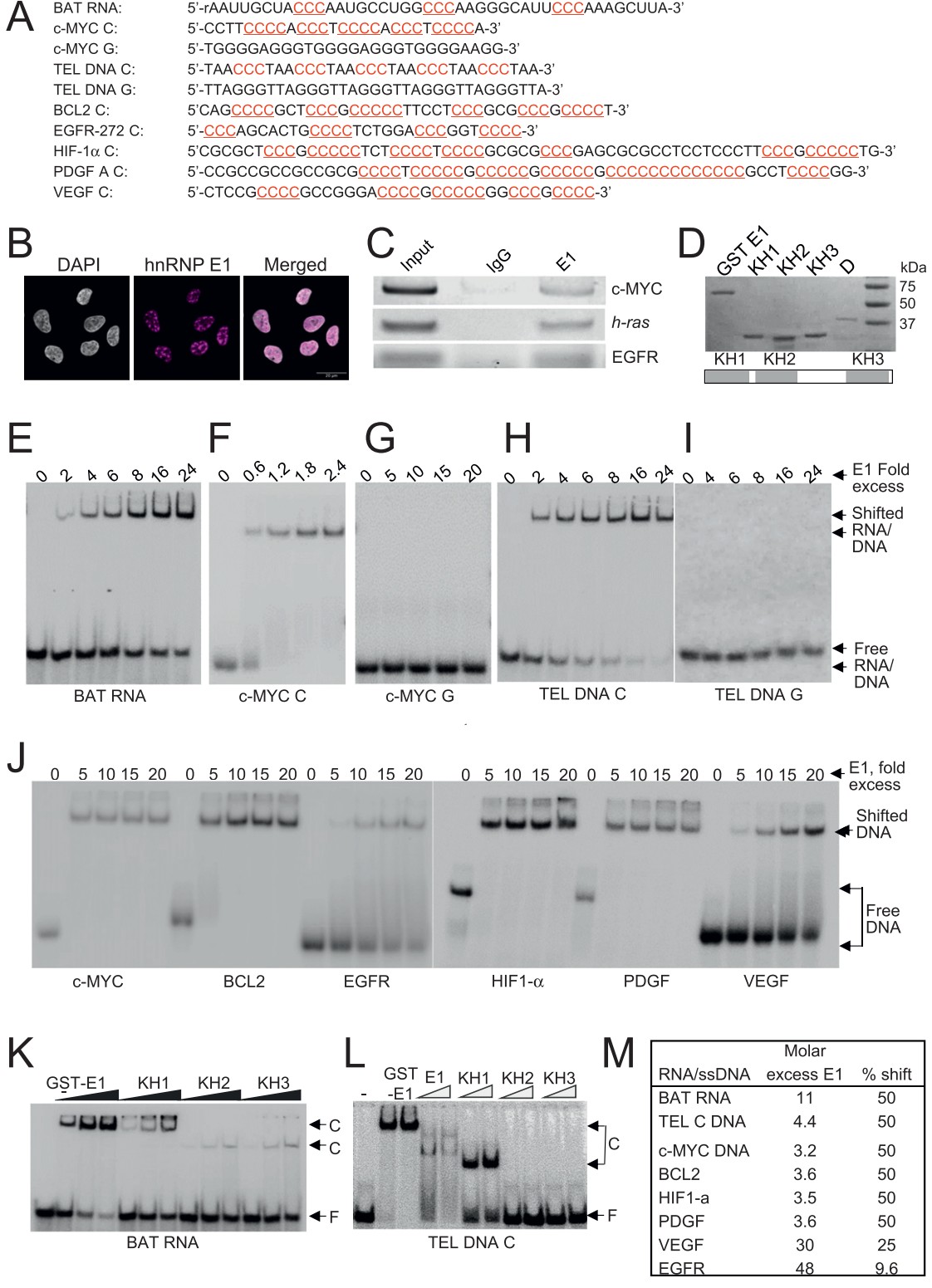

**Figure 1.   hnRNP E1 binds polycytosine-rich DNA tracts.**
**(A)** Oligonucleotides and deoxyoligonucleotides used in RNA electrophoretic mobility shift assay (REMSA)/electrophoretic mobility shift assay (EMSA). **(B)** Immunofluorescence of hnRNP E1 and DAPI staining showing localization of hnRNP E1. **(C)** ChIP-PCR products from A549 DNA showing enrichment of hnRNP E1 at promoter regions of *c-MYC*, *h-Ras*, and *EGFR* genes. **(D)** hnRNP E1 protein and its KH domains. **(D)** Top. SDS–PAGE gel showing GST fusion proteins of hnRNP E1; KH1, KH2, and KH3; and hnRNP D. Bottom. The three KH domains of hnRNP E1. **(E, F, G, H, I, J, K, L)** Autoradiograms of representative REMSA/EMSA assays. **(E)** REMSA showing hnRNP E1 binding to BAT RNA. **(F, G, H, I, J)** EMSA showing hnRNP E1 binding to *c-MYC* C (F), *c-MYC* G (G), TEL DNA C (H), TEL DNA G (I), and *c-MYC* C, *BCL2*, *EGFR*, *HIF1-α*, *PDGF*, and *VEGF* (J). **(K)** EMSA showing binding of hnRNP E1, KH1, KH2, and KH3 to BAT RNA. **(L)** EMSA showing binding of hnRNP E1, KH1, KH2, and KH3 to TEL DNA C. **(M)** Quantification of REMSA and EMSA assay.

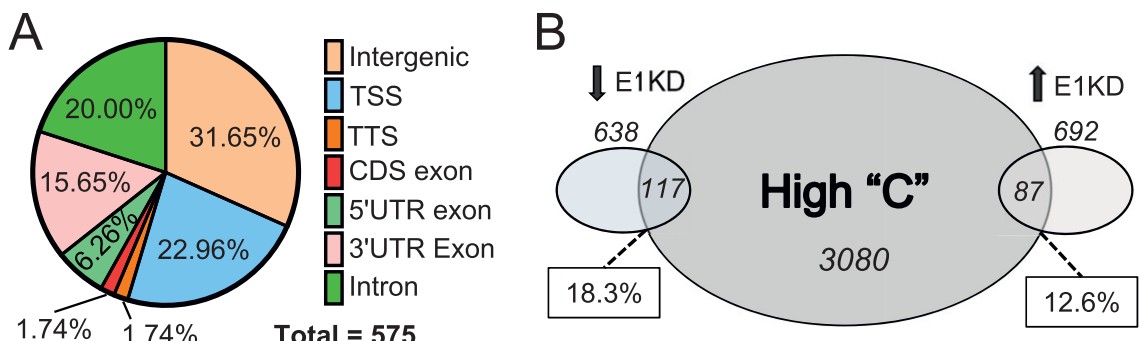

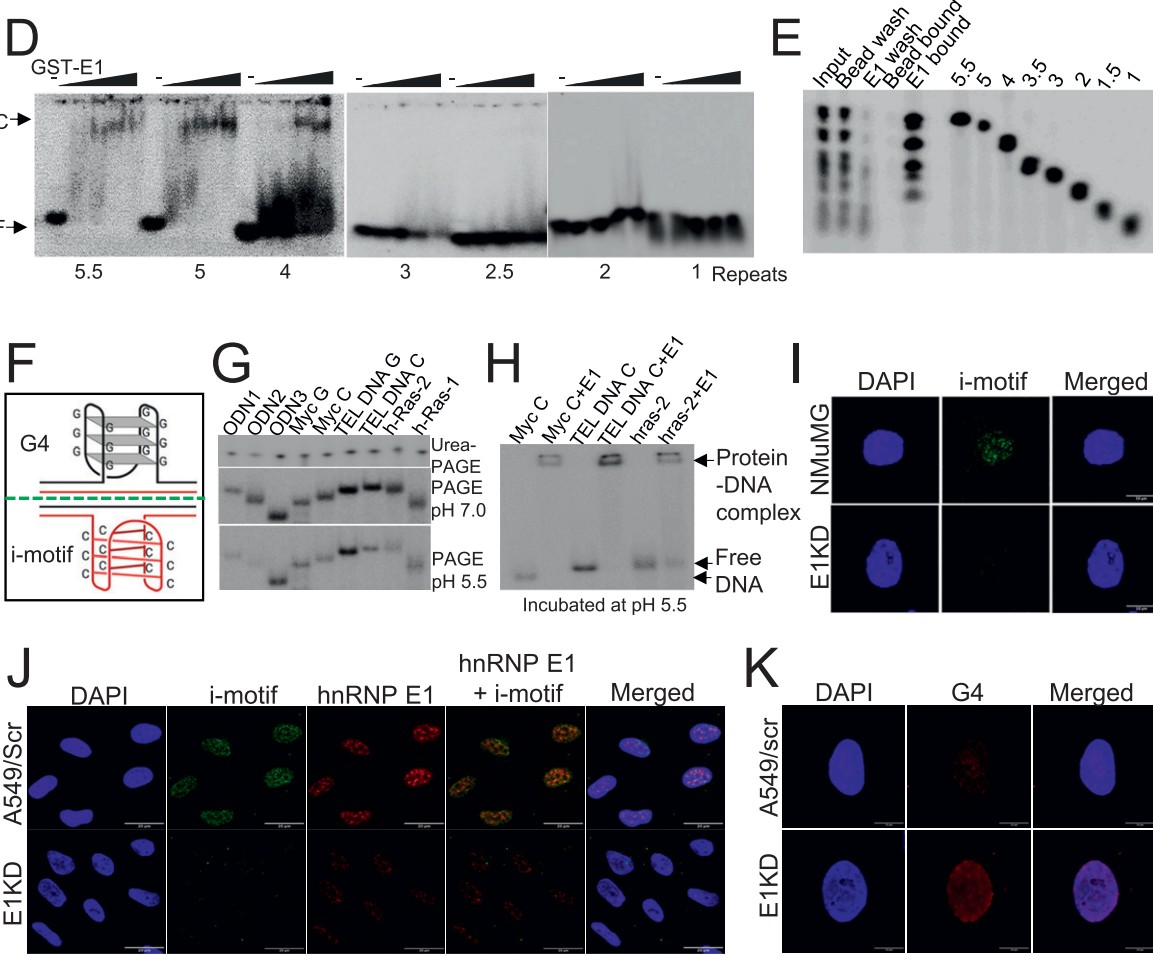

**Figure 2. Genome-wide DNA binding by hnRNP E1 and its cellular function.**
**(A)** Per cent of regions in the mouse (NMuMG) genome with at least 50% of "C" nucleotide. **(B)** Per cent (and number) of promoter proximal regions containing high-C sequences. **(C)** Two poly-C representative sequences from Table S1. **(D)** Autoradiographs of electrophoretic mobility shift assay showing differential binding of telomeric oligodeoxynucleotides containing one to five repeats of "CCCTAA" to GST-hnRNP E1 protein. **(E)** PAGE gel showing binding of telomeric oligodeoxynucleotides containing one to five repeats of "CCCTAA" upon loading on GST-hnRNP E1 immobilized on agarose beads. **(F)** Model showing i-motif and G4 structures. **(G)** Autoradiograph of urea-PAGE and PAGE (pH 7.5 and 5.5) gels showing fractionation pattern of various oligonucleotides. **(H)** Autoradiograph of a PAGE gel (run at pH 7.5) showing binding of hnRNP E1 to *c-MYC*, TEL DNA C and hRas2 deoxynucleotides incubated at pH 5.5. **(I)** Immunofluorescence of NMuMG control and E1KD cells showing i-motif foci. **(J)** Immunofluorescence of A549 control and E1KD showing i-motif foci, hnRNP E1 foci, and their colocalization. **(K)** Immunofluorescence of A549 control and E1KD cells showing G4 foci.

strongly than KH2 and KH3. When TEL DNA C was examined, whereas KH1 bound to the ssDNA, KH2, and KH3 did not show any significant binding (Fig 1L). KH1 bound to TEL DNA C less efficiently in comparison to full length protein. It may be noted that hnRNP E1 protein without a GST tag showed two shifted bands in EMSA assay (Fig 1L); however, because of the smear generated in EMSA lanes and the two bands generated by the protein (suggesting that the protein binds to nucleic acid as a dimer), we used the GST fusion version for quantification of protein–DNA interactions. hnRNP E1 bound less efficiently to BAT RNA than to TEL DNA C (Figs 1 and S3A). We analyzed binding of hnRNP E1 protein and its three domains to TEL RNA C, an RNA containing exactly same sequence as TEL DNA C. We observed that only KH1 domain binds to the TEL RNA C, but with much lesser efficiency than the whole protein (Fig S3B). Importantly, hnRNP E1 binds to TEL RNA C as strongly as it binds to TEL DNA C (Fig S3C and D) suggesting that the number of poly-C tracts, their frequency, number of intervening sequences, and number of "C"s per track determine nucleic acid binding by hnRNP E1. More data are presented later to confirm these observations. These experiments establish that hnRNP E1 binds specifically to poly-C–rich ssDNA tracts, but not to poly-G tracts or double-stranded DNA.

## Mouse genome contains numerous potential hnRNP E1 binding sites

hnRNP E1, and its role in gene expression and translation have been studied extensively in the mouse cell line NMuMG (Chaudhury et al, 2010b; Hussey et al, 2012; Grelet et al, 2017; Howley & Howe, 2018, 2019). We conducted an in silico high-throughput DNA sequence analysis (see details in the Materials and Methods section) to identify potential hnRNP E1-binding poly-C sites in the genome of mouse using data from experiments with NMuMG (hnRNP E1 in mouse cells is identical to that of human cells). 50–75 nucleotide long DNA sequences with at least 50% cytosine content were identified with their locations from the genomic sequence (Table S1). Strikingly, a large majority of these sequences were intergenic (31.65%) followed by transcription start sites (22.96%), introns (20%), and 3′-UTRs (15.65%). Whereas 6.26% of 5′-UTRs contained "C"-rich sequences, 1.74% of each of coding sequence (CDS) exons and transcription terminator regions contained such sequences (Fig 2A). Approximately 12.6% of the up-regulated and 18.3% of the down-regulated transcripts in E1KD cells matched with promoter-proximal sequences (Fig 2B). All high "C" sites identified by this analysis do not contain tracts of poly-C; sequences that contain multiple "CCC" tracts, each with minimum three "C"s were identified and are highlighted in Table S1. Two such sequences containing poly-C tracts including telomeric sequences including the telomeric sequence from in silico analysis were identified and are shown in Fig 2C. A ChIP analysis with NMuMG cells using hnRNP E1 antibody showed that hnRNP E1 localizes to mouse telomeres (Fig S4A and B).

To further understand the mechanism of hnRNP E1 binding to DNA, we examined the number of poly-C tracts and number of cytosines per tract required for hnRNP E1 binding to a DNA sequence. DNA sequences at promoter proximal regions of oncogenes tested in Fig 1 contain variable number of tracts and variable number of "C"s per tract; in contrast, telomeres in human and mouse contain exact

repeats of "CCCTAA" with each repeat containing three "C"s. This repeat nature of telomeric sequence was used to dissect the hnRNP E1–DNA interaction. We conducted two different DNA binding assays with oligodeoxynucleotides containing various numbers of "CCCTAA" repeats. (1) EMSA assay with oligodeoxynucleotides containing one to five repeats of "CCCTAA" in the presence of hnRNP E1 protein showed visible shifts of DNA containing four and five repeats of "CCCTAA," but smears appeared with DNA containing two to three repeats of "CCCTAA" at high concentrations of the protein; no binding was visible with a single repeat of "CCCTAA" (Fig 2D). (2) We analyzed hnRNP E1–DNA interactions by a second method in which hnRNP E1 protein was immobilized on agarose beads, a mixture of oligodeoxynucleotides containing various repeats of "CCCTAA" was loaded, and the bound/unbound fractions were identified. As shown (Fig 2E), DNA containing two and more repeats of "CCCTAA" bound to hnRNP E1. The second method suggests that minimum two repeats of "CCCTAA" is necessary for binding to hnRNP E1. In a different experiment we determined the number of "C"s required per tract for hnRNP E1 binding using telomeric oligodeoxynucleotides that contain five tracts of "CCC" and their mutant versions. Mutation of single first "C"s in "CCC" tracts did not affect hnRNP E1 binding to DNA (data not shown). Simultaneous mutations of four of five first "C"s affected hnRNP E1 binding adversely, but did not completely abolish binding; however, simultaneous mutations of all five first "C"s completely abolished hnRNP E1 binding to DNA (Fig S4C). These data show that the number of poly-C tracts in a DNA sequence and number of "C"s per track determine hnRNP E1–DNA interactions.

## hnRNP E1 regulates DNA secondary structures

The poly-C tracts are spread throughout the genome including at promoters, inside genes, at UTRs and in non-transcribed parts of genome. What is common to poly-C tracts at these different sites? Because poly-C tracts can form secondary structures such as i-motifs (Miglietta et al, 2015; Wolski et al, 2019) (Fig 2F), we wanted to know if hnRNP E1 binds to their i-motif structures. Various Poly-C–rich oligodeoxynucleotides that have already been shown to generate i-motifs (Phan & Leroy, 2000; Dai et al, 2010; Reilly et al, 2014; Miglietta et al, 2015) were fractionated in denaturing or non-denaturing polyacrylamide gels. As shown in Fig 2G, whereas all the ssDNAs tested fractionated according their size in the urea denaturing gel, they fractionated according to their secondary structures in non-denaturing gels either at pH 5.5 or at 7.5. Three poly-C containing sequences, namely, c-MYC, TEL DNA C, and hRas2 oligonucleotides were incubated at pH 5.5 with or without hnRNP E1 protein, fractionated by non-denaturing PAGE pH 7.5, and analyzed further. As shown (Fig 2H), hnRNP E1 protein bound to c-MYC C, TEL DNA C, and hRas-2Y DNA.

We explored a possible role for hnRNP E1 in i-motifs in the cell. By immunofluorescence analysis using an i-motif (iMab scFv) antibody, discrete i-motif foci were observed in the DAPI-stained regions in both wild type NMuMG cells (Fig 2I) and A549 cells (Figs 2J and S4D). In contrast to wild type cells, E1KD cells displayed significant reduction in i-motif signal in both cell lines Figs 2I and J and S4D. A significant fraction of the i-motifs colocalized with hnRNP E1 foci in A549 cells (Fig 2J) suggesting that hnRNP E1 might play a role in the formation or maintenance of i-motif in the cell. Because

poly-G–rich tracts are complementary to poly-C tracts in the genome, we examined any changes to poly-G tracts. Because poly-G tracts are known to form G4s, we explored the status of G4s in A549 cells and their E1KD derivative by immunofluorescence using a G4-specific antibody. G4 accumulation was limited in A549 control cells but showed significant increase in E1KD cells (Fig 2K). All these data showed that hnRNP E1 may play a role at poly-C/poly-G tracts in the genome by regulating the competing formation of i-motif and G4s.

### hnRNP E1 knockdown induces DNA damage signaling

Because hnRNP E1 binds to poly-C tracts spread throughout the genome and because hnRNP E1-knockdown cells display increase in G4s, which are sites for DNA damage, we wanted to know if DNA damage occurs in E1KD cells. Phosphorylation of histone H2AX at Ser139 (γ-H2AX) is one of the first steps of DDR (Kuo & Yang, 2008). Immunofluorescence analysis of γ-H2AX in A549 and its E1KD derivative showed a twofold increase in γ-H2AX accumulation in E1KD cells in comparison to control cells (Fig 3A and B). Western blot analysis also showed an increase in γ-H2AX in E1KD cells in comparison to control cells (Fig 3C); data in later sections also show similar results (Fig 4D, E, and G). Accumulation of G4 structures can block DNA replication and activate a DDR that involves checkpoint activation (León-Ortiz et al, 2014). Since hnRNP E1-knockdown cells displayed G4 accumulation, we predicted DNA damage and checkpoint activation in these cells. We analyzed γ-H2AX and G4s separately and for their possible colocalization by immunofluorescence. Both γ-H2AX and G4s foci showed increase in E1KD cells over control cells, and there was substantial colocalization of γ-H2AX foci with G4 foci (Fig 3D). These data strongly suggest basal DNA damage in E1KD cells. We explored the possibility of DNA

damage near the binding sites of hnRNP E1 in its absence. HU and UV can cause DNA damage and thus activate DDR (Oh et al, 2011). Cells with or without the presence of hnRNP E1 were treated with HU and activation of histone H2AX (γ-H2AX) was analyzed at hnRNP E1 binding sites by ChIP-PCR using γ-H2AX antibody. As shown (Fig 3E), localization of γ-H2AX at c-MYC promoter was found to be enhanced in E1KD cells in comparison to scrambled cells especially upon HU treatment. These results strongly suggest a protection role for hnRNP E1 at its binding sites.

### Genotoxin treatment exacerbates DNA damage signaling in E1KD cells

A549 and its E1KD derivative cells were exposed to HU and UV. Analysis of the cell cycle distribution pattern revealed that E1KD cells contained a small increase in G2 cell population and decrease in G1 cells from control cells; UV treatment significantly increased the G2 population and decreased G1 population in E1KD cells (Fig 4A and B). The altered cell cycle distribution reflects G2/M checkpoint activation in E1KD cells that is enhanced by UV. Upon UV exposure, E1KD cells displayed increase in cyclopyrimidine dimer (CPD) formation in comparison to control cells (Fig 4C) suggesting increased DNA damage. UV treatment also caused increased signals of checkpoint activation in E1KD cells in comparison to control (A549 and NMuMG) cells as evidenced by increased activation of γ-H2AX, pATM, pATR, p53, and phospho-p53 (Fig 4D and E). PCNA is the DNA clamp that is loaded to DNA by replication factor C complex, in coordination with RPA and Fen1, PCNA loads DNA polymerase δ at millions of Okazaki fragments synthesized during DNA replication, and loads repair DNA polymerases at DNA damage sites (Fox et al, 2011; Zhang et al, 2011; Boehm et al, 2016). UV and HU treatment of

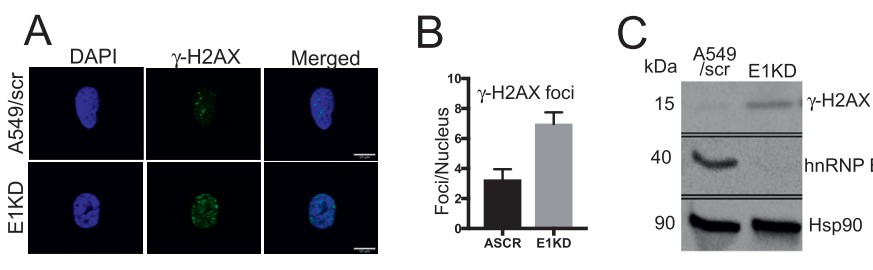

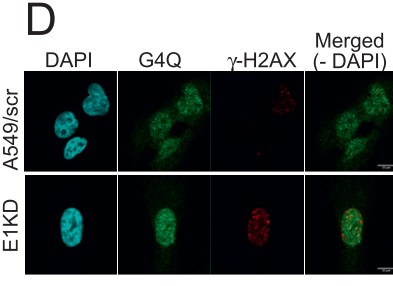

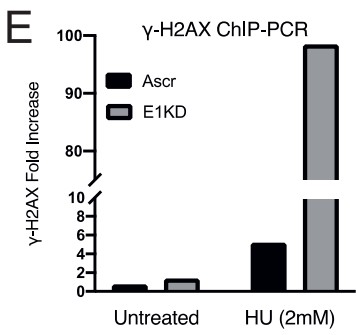

**Figure 3. Role of hnRNP E1 in genome integrity.**
**(A, B)** Immunofluorescence showing γ-H2AX activation in E1KD cells. **(C)** Western blot showing γ-H2AX activation in E1KD cells. **(D)** Immunofluorescence showing increase in and colocalization of γ-H2AX and G4 foci in E1KD cells. **(E)** ChIP-qPCR analysis showing γ-H2AX localization at c-MYC promoter of scrambled (Ascr) and hnRNP E1 knockdown (E1KD) cells.

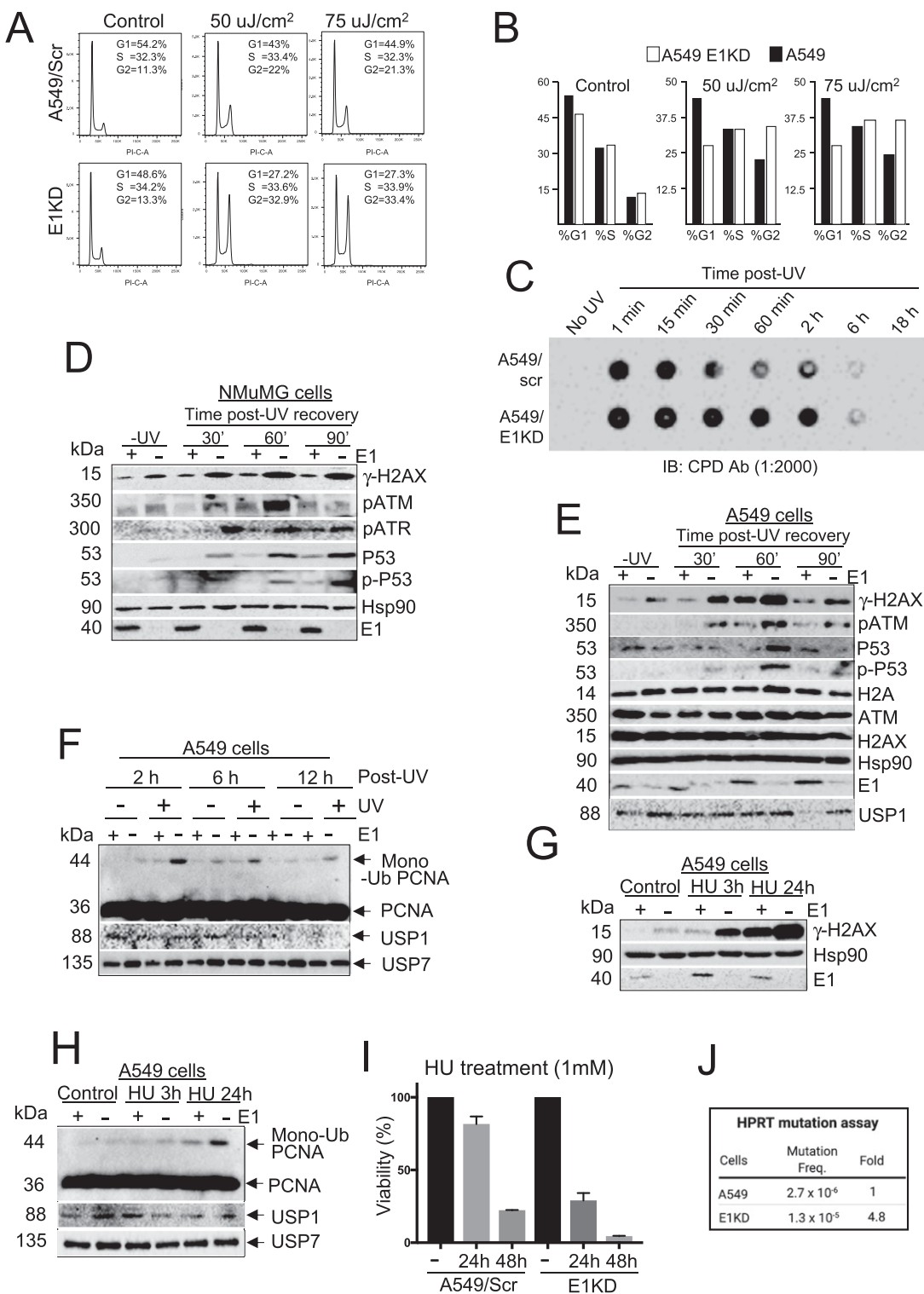

**Figure 4. E1KD phenotype acts synergistically with HU/UV and promotes mutation.**
**(A, B)** Cell cycle distribution of A549 control and E1KD cells exposed to UV. **(C)** Blot showing CPD incorporation in wild-type and E1KD cells. **(D, E)** Western blots showing effects of UV on checkpoint proteins in A549 control and E1KD (D) cells, and checkpoint proteins in NMuMG control and E1KD cells (E). **(F)** Western blot showing proliferating cell nuclear antigen monoubiquitination, Usp1, and Usp7 in A549 control and E1KD cells recovering from UV exposure (h, hours). **(G, H)** Western blots showing γ–H2AX activation (G) and proliferating cell nuclear antigen monoubiquitination along with Usp1 and Usp7 (H) in A549 control and E1KD cells treated with HU. **(I)** Histogram of clonogenic assay with A549 control and E1KD cells exposed to HU. **(J)** Mutation frequency in A549 control and E1KD cells.

cells cause PCNA monoubiquitination in parallel with disappearance of deubiquitinating enzyme Usp1 and Usp7 (Niimi et al, 2008; Brown et al, 2009; Zlatanou et al, 2016). A549 and E1KD cells were exposed to UV, allowed to recover, and harvested at different time points. As shown in Fig 4F, UV exposure increased PCNA monoubiquitination in E1KD cells than control cells that lingered for more than 12 h.

HU treatment reduces the cellular dNTP pool, slowing the rate of DNA synthesis and generating ssDNA (Singh & Xu, 2016). ssDNA generated at poly-G tracts during DNA repair and replication form G4s that can block the advancing replication machinery (Papadopoulou et al, 2015). To investigate whether replication stress exacerbates the defects associated with E1KD, we treated cells with HU and observed increased γ-H2AX formation in E1KD cells in comparison to control cells over the basal γ-H2AX activation (Fig 4G). Exposure to HU induces PCNA monoubiquitination, which facilitates its DNA repair functions (Niimi et al, 2008). Upon HU treatment, PCNA monoubiquitination increased in E1KD cells compared to control cells (Fig 4H) and PCNA monoubiquitination was high despite the presence of the deubiquitinating enzymes Usp1 and Usp7 (Fig 4H). HU treatment also caused increased cell death (Fig 4I). Thus, reduction in dNTPs by HU acts synergistically with the E1KD phenotype to induce enhanced DNA damage signaling.

### hnRNP E1 suppresses mutation

The basal DNA damage signals associated with the E1KD phenotype, including G4 accumulation, increased damage signaling including γ-H2AX activation, and RPA accumulation, are all associated with genomic instability. DNA damage leads to PCNA monoubiquitination, which in coordination with RPA and Fen1 replaces DNA polymerase δ with mutagenic or non-mutagenic DNA polymerases (Fox et al, 2011). Because hnRNP E1 functionally interacts with RPA and Fen1 and because hnRNP E1-knockdown cells displayed PCNA monoubiquitination, we predicted that E1KD cells may be prone to spontaneous mutations. In a hypoxanthine phosphorybosyl transferase (HPRT) gene assay that detects mutations (Johnson, 2012) (See the Materials and Methods section for details), E1KD cells showed an ~fivefold increase in mutation frequency over control cells (Fig 4J), indicating that hnRNP E1 plays a critical role in maintaining genome integrity.

## Discussion

To our knowledge, this is the first report on genome-wide binding of hnRNP E1 to polycytosine-repeats and its global role in maintenance of DNA secondary structures such as i-motifs and suppression of G4s, and protection of cells from DNA damage/replication stress. hnRNP E1 knockdown resulted in increased accumulation of γ-H2AX, G4s, RPA, PCNA monoubiquitination, and increase in mutations. This work suggests that the genome-wide poly-C DNA binding is one of the ways by which hnRNP E1 monitors genome integrity.

Various hnRNPs bind poly-G or poly-C tracts or both (Herbert, 2020). All these studies have focused on individual promoters or telomeres. In the present study we have observed that hnRNP E1 binds exclusively to poly-C tracts and the i-motif structures of several promoter regions and telomeres, but not to their complementary poly-G tracts; we have also identified numerous poly-C

tract-containing potential hnRNP E1 binding sites in mouse genome. Our analysis identified several mouse promoters that contain potential hnRNP E1 binding poly-C tracts and human gene promoters to which hnRNP E1 binds. Our finding that hnRNP E1 binds to poly-C tracts present at promoter proximal regions of several oncogenes and to telomeres suggests that the human genome also contains numerous hnRNP E1 binding poly-C tracts. In addition, although the consensus BAT RNA element contains poly-C tracts (Brown et al, 2016), hnRNP E1 binds to BAT RNA elements which contain poly-pyrimidine tracts which are not poly-C (Chaudhury et al, 2010b; Brown et al, 2016). We predict that hnRNP E1 not only binds to poly-C tracts but also to other sequences in the genome; future genome-wide studies will reveal all hnRNP E1 binding sites in the genome as well as the sites that are damaged in the absence of hnRNP E1 with or without genotoxin treatment. All potential binding sites may not be available for hnRNP E1 binding in a cell at a given time; however, the sequences should be available for hnRNP E1 binding during DNA replication, repair, transcription elongation/termination when DNA: RNA hybrids are generated, or in a tissue-specific manner.

hnRNP E1 binds to i-motifs formed by poly-C tracts, maintains i-motifs, and suppresses G4s in the cell. Recently it has been observed that i-motifs and G4s are mutually exclusive (King et al, 2020). G4s can inhibit DNA replication, but various proteins including helicases, Fen1, and hnRNPs resolve such structures. During DNA replication and post-replication repair PCNA, in collaboration with RPA and Fen1, loads DNA polymerases that can be mutagenic or non-mutagenic (Fox et al, 2011; Zhang et al, 2011; Boehm et al, 2016). This suggests that hnRNP E1 may help DNA replication/repair proteins at G4s and other damage sites in poly-C tracts. Interaction of hnRNP E1 with these proteins and other proteins may help facilitate replication fork movement past the polyguanine/polycytosine repeats.

PCNA encircles DNA, helps load various DNA polymerases on DNA, and regulates DNA replication and repair (Boehm et al, 2016). When DNA replication machinery encounters a lesion, PCNA is ubiquitinated (Zhang et al, 2011). Monoubiquitinated PCNA recruits error-prone DNA polymerases and leads to translesion DNA synthesis (TLS), whereas polyubiquitinated PCNA recruits error-free polymerases causing template switch and repair by homologous recombination proteins (Lee & Myung, 2008). Exposure to genotoxic agents such as UV, HU, methyl methanesulfonate (MMS) or $H_2O_2$ induces PCNA monoubiquitination (Niimi et al, 2008). Whereas Rad6/Rad18 E2 conjugate/E3 ligase cause PCNA monoubiquitination, Usp1 and Usp7 cause PCNA deubiquitination with some difference in case of HU (Niimi et al, 2008; Fox et al, 2011). We observed increased PCNA monoubiquitination in E1KD cells upon HU and UV treatment that persists for a long time. We predict that extensive DNA damage occurring in E1KD cells are repaired slowly because hnRNP E1 is not available at damaged poly-C sites to facilitate loading of RPA, Fen1, DNA polymerases, and other repair proteins at these sites.

Our work shows that hnRNP E1 binds to poly-C tracts genome-wide and monitors genome integrity. We propose (Fig 5) that, in control cells, DNA damage-induced activation of checkpoints and repair proceeds normally; hnRNP E1 present at poly-C sites cooperates with DNA damage sensors and other checkpoint proteins in signaling and repair. In the absence of hnRNP E1, repair at poly-C sites slows down because signaling and loading of repair proteins is slow; however, other proteins such as RPA carry out the functions of hnRNP E1 directly at these

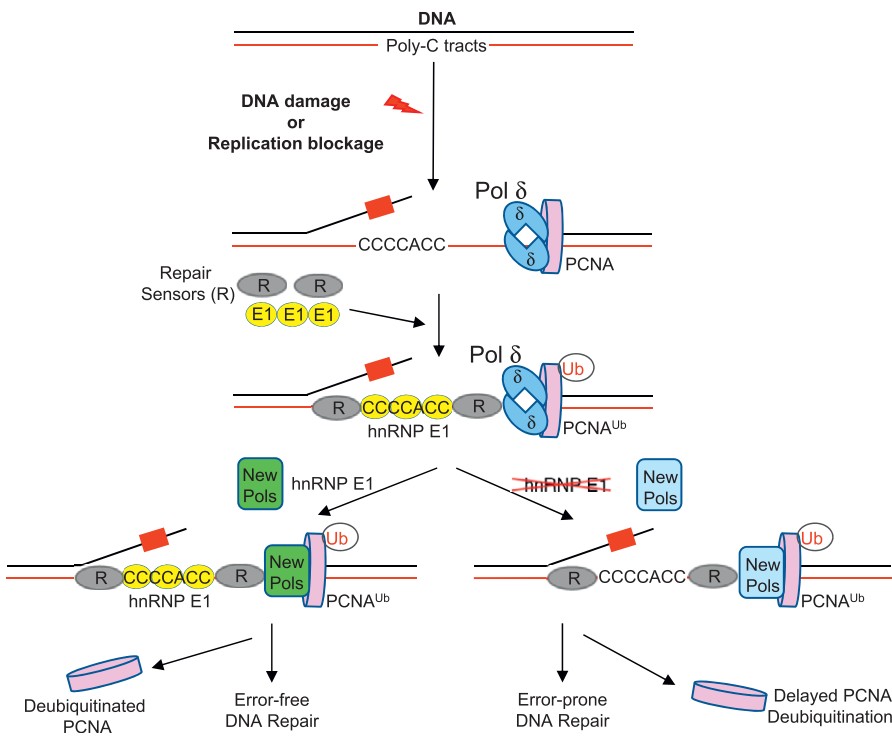

**Figure 5. Model showing the role of hnRNP E1 in genome integrity.**
Upon DNA damage, hnRNP E1 bound at polycytosine tracts activates damage signaling and repair with the loading and assistance of RPA (R). Proliferating cell nuclear antigen (PCNA) is monoubiquitinated, replaces DNA polymerase $\delta$ with TLS polymerases, which are often error-free, and finally PCNA is deubiquitinated. When DNA damage occurs in the absence of hnRNP E1, only RPA functions at polycytosine tracts albeit slowly, PCNA monoubiquitination is enhanced and lingers, error-prone TLS polymerases replace DNA polymerase $\delta$, and signaling and repair are attenuated.

sites in DNA repair. While the manuscript was being revised a new publication showed that the iron-binding and DNA damage functions are separable from RNA binding functions (Patel et al, 2021); the authors also suggested that the nucleic acid and iron binding by hnRNP E1 might enhance the assembly or repair of iron cofactors in DNA- and RNA-modifying enzymes. Although this is a major possibility in untreated E1KD cells, we predict the existence of additional possibilities; in addition, sensing and repair of genotoxin-induced DNA damage may involve different mechanism. Although we have identified one specific site of DNA damage in the current work, it is possible that DNA damage in the absence of hnRNP E1 can occur both at hnRNP E1–binding sites and at other nonspecific sites; genome-wide work is underway to identify DNA damage sites in E1KD cells. The E1KD phenotype acts synergistically with genotoxic stress by HU and UV. hnRNP E1 binds to poly-C tracts throughout the genome upon their availability, controlling DNA secondary structures to protect genomic integrity. Noncanonical DNA structures such as G4s and i-motifs have become important to study both chemically and biologically because of their roles in cancer, neurological disorders, and other hereditary diseases. Biophysical studies will elucidate the mechanisms by which hnRNP E1 binds to DNA, and maintains and regulates secondary structures. Future studies will provide new insight into the role hnRNP E1 plays in genome maintenance and cancer development.

# Materials and Methods

### Cell cultures, antibodies, and plasmids

Human lung epithelial cell line A549 (ATCC), its hnRNP E1 knockdown (A549/E1KD or E1KD), normal mouse mammary epithelial (NMuMG) cell line, and its hnRNP E1 knockdown (NMuMG/E1KD or E1KD) derivative cells (Hussey et al, 2011, 2012) were cultured in DMEM supplemented with 5% FBS and 5% FCS at 37°C in a 5% $CO_2$ humidified chamber. Antibodies used in the work were against the following proteins: hnRNP E1 (mouse; Abnova; mouse, MBL; and rabbit, raised against purified hnRNP E1), TRF2 (rabbit, NB10056506; Novus), Hsp90 (mouse; SCBT), FLAG (rabbit; Sigma-Aldrich), 6XHis (rabbit; Cell signaling), phospho-ATM (MAB 2401; Abnova), ATM (2873S; Cell signaling), phospho-ATR, ATR (13934S; Cell signaling), phospho-p53 (9284S; Cell signaling), p53 (2524; Cell signaling), γ-H2A.X (rabbit, 9718S; Cell signaling), Histone H2A.X (7631T; Cell signaling), PCNA (mouse, Cat no. 2586S; SCBT), Usp1 (rabbit, 8033S; Cell signaling), Usp7 (mouse, HAUSP; SCBT), RPA32 (rat, 2208S; Cell signaling), Rad6 (R6A/R6B, rabbit, 4944S; Cell signaling), Rad18 (rabbit, 9040S; Cell signaling), and Fen1 (rabbit; SCBT). Plasmids containing hnRNP E1 and its three KH domains namely KH1, KH2, and KH3 have been described previously (Chaudhury et al, 2010b; Brown et al, 2016); each of the four ORFs contained an N-terminal GST tag.

### Proteins

Plasmids containing GST-fusion were grown in *Escherichia coli* strain DH5α or BL21DE3 RIPL. Overnight cultures were inoculated into fresh Luria broth (LB) containing ampicillin (50 $\mu$g/ml) and grown to $A_{600}$ 0.5. The proteins were induced with 100 $\mu$M IPTG for 3 h at 37°C. Cells were harvested, suspended in buffer A containing 25 mM Tris–HCl (pH 7.6), 1 mM EDTA (pH 8), 10% sucrose, 10 mM $\beta$ mercaptoethanol ($\beta$ME), and lysozyme (1 mg/ml) and frozen at −80°C. Frozen cells were lysed by thawing on ice. The cell lysate was

centrifuged at 27,000g in Sorvall SS-34 rotor for 1 h at 4°C. The supernatant was mixed with glutathione sepaharose beads (GE) equilibrated with buffer B (25 mM Tris–HCl (pH 7.6), 1 mM EDTA (pH 8), 10 mM βME, 150 mM NaCl, and 2% glycerol). Binding of proteins to the beads was carried out at 4°C for 2 h. The beads were centrifuged, washed three to five times with buffer B, and proteins were eluted from the beads with buffer B containing 10 mM reduced glutathione (pH adjusted to 7.6 with 1 M Tris). The proteins were adjusted to 50% glycerol in buffer B and kept frozen at –80°C.

## SDS gel electrophoresis and immunoblotting

All cell extracts (cellular, nuclear and cytoplasmic extracts) or purified proteins were fractionated by SDS–PAGE, stained with Coomassie blue or transferred to PVDF membranes (Bio-Rad). Proteins on PVDF membranes were probed with appropriate primary antibodies (mostly at a dilution of 1:1,000) and secondary antibody–HRP conjugates (1:5,000 dilutions). The Western blots were developed using ECL and analyzed using a Bio-Rad ChemiDoc Gel Imaging System.

## Oligonucleotides and deoxyoligonucleotides

All the deoxyoligonucleotides and (oxy)oligonucleotides are described in Table S2.

## EMSAs and RNA EMSA (REMSA)

Oligonucleotides and deoxyoligonucleotides were 5′-end labeled in a 50 μl reaction mix that contained 50 pmol of an oligodeoxynucleotide, 1× NEB T4 PNK buffer (New England Biolabs), 150 μCi γ-$^{32}$P ATP (specific activity, 6,000 Ci/mmole) and 10 U of T4 polynucleotide kinase, and the samples were incubated at 37°C for 1 h. The labeled oligodeoxynucleotide were purified through Sephadex G25 (GE Healthcare) spin columns. $^{32}$P-labeled oligoribonucleotides or oligodeoxynucleotide (20 fmol) were mixed with various concentrations of proteins in 50 μl reaction buffer that contained 25 mM Tris–HCl (pH 7.6), 1 mM EDTA, 2% glycerol, 10 mM βME, 50 μg/ml BSA, and with or without 1.5 ng heat denatured (sheared) *E. coli* genomic DNA. The reaction mix was incubated at RT for 15 min before being mixed with loading dye containing 5% glycerol and bromophenol blue. The samples were loaded on an 8–10% polyacrylamide gel in 1× Tris-Borate-EDTA (TBE) buffer run at 150 V for 90 min. The gels were dried and exposed to a PhosphoImager screen and analyzed by a Typhoon FLA 1900 PhosphoImager. EMSA bands were analyzed and quantified using ImageJ.

## Chromatin immunoprecipitation (ChIP) and PCR

Formaldehyde crosslinking and ChIP were carried out as below. A549/scrambled and E1KD cells were grown in DMEM medium containing puromycin. NMuMG/E1KD cells expressing the FLAG-tagged hnRNP E1 (Chaudhury et al, 2010b) were grown in DMEM medium containing puromycin and G418 (200 μg/ml) to 70% confluency. Cells were washed, scraped, and collected in PBS. Cells were cross-linked with 1% formaldehyde on ice for 15 min after which glycine was added to 125 mM. After 15 min, cells were washed twice in 10 ml PBS and pellets were frozen. The frozen cells were suspended in ChIP lysis buffer (25 mM Tris–HCl, pH 7.6, 1 mM EDTA, 150 mM NaCl, 1% sodium deoxycholate, 0.1% SDS, and protease inhibitor; Roche) and lysed by sonication (6 × 30 s pulses with 2 min intervals on ice or in an automatic sonicator for 20 min). The lysates were cleared by centrifugation. 1/10$^{th}$ of the lysate was collected and used as input. For experiments on telomeres (NMuMG cells), the remaining lysate was made up to 3 ml and divided into three equal volumes of 1 ml each, to which IgG control, anti-FLAG antibody (Sigma-Aldrich), or anti-Trf2 antibody (Novus) was added. Similarly, for A549 derivative, 1/10$^{th}$ volume was collected as input and the remaining volume was divided into two parts: to one IgG was added and to the other hnRNP E1 antibody was added. The tubes were rotated on a Rototorque overnight at 4°C. The next day, protein A agarose (100 μl of 50% slurry) was added and the tubes were rotated for 2 h at 4°C. The beads were washed 3 × 1 ml of ChIP lysis buffer without SDS, 1 × 1 ml ChIP lysis buffer without detergent, and finally suspended in 100 μl of the same buffer. Crosslinking was reversed by treating samples (including input samples) at 65°C overnight. The samples were processed by a gel extraction kit system (Thermo Fisher Scientific) and DNA was dissolved in 30 μl of 10 mM Tris-1 mM EDTA (TE) each.

Telomeric PCR was conducted as described (Cawthon, 2002, 2009). Semi-quantitative PCR was performed using Maxima Hot Start Taq polymerase (Thermo Fisher Scientific), 100 nM telomere primers tel1 and tel2 (Table S2) with the following conditions; 95°C 10 min, 40 cycles of 95°C 15 s and 58°C 2 min. PCR end products were visualized on 2% agarose gels stained with ethidium bromide and imaged using the Bio-Rad ChemiDoc system. Real-time PCR was performed using iQ SYBR Green Supermix (Bio-Rad) and 100 nM telomere primers as per the manufacturer's instructions. Ct values were obtained using the CFX384 Real-Time System (Bio-Rad) and the following cycling conditions; 95°C 10 min, 40 cycles of 95°C 15 s and 58°C 2 min. Primers used for both real-time and semi-quantitative PCR for telomeres were designed according to Cawthon (2009) and primer sequences are shown in Table S2. For PCR of promoter proximal regions in A549 cells, the PCR conditions were: initial denaturing 95°C - 5 min followed by 32 cycles of 95°C—30 s, 60°C—30 s, and 72°C—30 s, followed by final extension at 72°C—5 min.

## Flow cytometry

Cells were grown in DMEM medium with appropriate antibiotics, irradiated with UV (as specified in the text) and were further grown for appropriate time points. Cells were washed with 2 × 10 ml PBS and collected from plates after trypsinization. Cells were centrifuged and finally suspended in water to a final concentration of 10$^6$/ml. Cells were treated with RNase and finally ethanol was added to a final concentration of 70%. Before flow cytometry analysis, cells were centrifuged, ethanol was removed, and cells were suspended in water. Propidium iodide (Sigma-Aldrich) was added and cells were analyzed by an LSRFortessa/X20.

## Immunofluorescence

Cells were grown in DMEM with appropriate antibiotics on coverslips in six-well plates and treated with appropriate agents. Cells were processed for immunofluorescence as described (Zeraati et al, 2018) with minor modifications. Cells were fixed by first adding

equal volume of 2% paraformaldehyde in PBS by incubating for 2 min at room temperature. Medium with paraformaldehyde was replaced with PBS containing 1% paraformaldehyde and the plates were incubated at 4°C for additional 10 min. Cells were then treated with PBS + 0.1% Triton X-100 for 30 min at 4°C. Cells were blocked overnight with 1 ml of SuperBlock (Thermo Fisher Scientific). After blocking, the cells were incubated overnight with primary antibody in SuperBlock at 4°C. Coverslips were washed with 3 × 1 ml PBS + Tween 20 (0.1%) and incubated with secondary antibody tagged with Alexa Fluor (488 or 568). After incubation for 1 h, coverslips were washed with 3 × 1 ml PBS + Tween 20 (0.1%) and mounted with DAPI (Thermo Fisher Scientific). Cells were analyzed by a Nikon confocal microscope.

### Genomic DNA preparation and immune-dot blot assay

Genomic DNA preparation after UV exposure and immune-dot blot assay were conducted essentially as described previously (Choi et al, 2015). Briefly, the cells were harvested at different times post-UV recovery and were resuspended in 400 $\mu$l buffer P1 (10 mM Tris-Cl, [pH 8.0], 1 mM EDTA, and 100 $\mu$g/ml RNase A). Cells were lysed by adding 55 $\mu$l of 10% SDS and incubated for 15 min at room temperature. 140 $\mu$l of 5 M NaCl was added, and the tubes were inverted gently 10 times and stored overnight at 4°C; alternatively, cells were lysed by adding Triton X-100 to 1% and keeping the lysates on ice for 15 min. The lysate was centrifuged in an Eppendorf microcentrifuge at maximum speed (20,000$g$) for 1 h at 4°C. The insoluble pellet was processed for preparation of genomic DNA whereas the soluble fraction was processed for preparation of soluble, free DNA (if needed). The pellet was resuspended in a 10× packed cell volume of PBS. The resuspended pellets were treated with 20 $\mu$g/ml of RNase A at 37°C for 20 min followed by proteinase K (0.25 $\mu$g/ml in 0.5% SDS) treatment for 1 h at 50°C. The genomic DNA was extracted first with phenol/chloroform, and then with chloroform. DNA was precipitated with 1/10 volume of 3 M sodium acetate and 2.2 volume of ethanol. The pellet was washed with 500 $\mu$l of 70% ethanol and resuspended in 200 $\mu$l of TE, and total DNA amount was determined by NanoDrop.

Purified genomic DNA was heat-denatured, chilled on ice, and then loaded onto a nitrocellulose membrane by a dot blot apparatus. The membrane was baked at 80°C for 30–60 min and probed with anti-CPD mouse antibody (at 1:1,000 dilution; as described for Western blotting method). After incubation with HRP-conjugated anti-mouse secondary antibody (1:5,000), chemiluminescent signals were detected with ECL reagent (GE Healthcare Life Sciences) using a Bio-Rad Imager.

### Clonogenic assay and trypan blue staining

Clonogenic assay was carried out as described (Rafehi et al, 2011) with minor modification as below. Cells were seeded at $5 × 10^5$/plate on 10 cm plates, grown for 2–3 d and then treated with different concentrations of HU for appropriate times. Cells floating in the medium and the attached cells (upon trypsinization) were pooled. Cells were centrifuged, suspended in DMEM medium, and counted with trypan blue. 300–500 cells were plated on 6 cm plates and incubated for 10–14 d. The colonies were fixed with 10% formaldehyde for 15 min and were stained with crystal violet in 50% methanol for 30–60 min. The plates were washed with water and colonies counted after taking photographs.

### Mutation frequency analysis using HPRT assay

Mutation frequency was determined using hypoxanthine phosphorybosyl transferase (HPRT) assay (Johnson, 2012). Cells were grown in DMEM medium containing hypoxanthine–aminopterin–thymidine (HAT; Sigma-Aldrich) medium for 5 d to eliminate pre-existing HPRT mutants. After HAT treatment, the cells were grown in DMEM containing hypoxanthine–thymidine (Sigma-Aldrich) supplement so that both the de novo nucleotide biosynthesis pathway and the salvage pathway were able to function. Cells were harvested and $5 × 10^5$ cells were plated on 10 cm dishes containing DMEM. After 2 d, cells were irradiated with UV (50 or 75 $\mu$J/cm$^2$) and grown for 2 d. Cells were harvested and $10^6$ cells were plated on 10 cm dishes containing DMEM and 6-thioguanine (6-TG) to select HPRT mutants; HPRT$^+$ cells incorporate 6-TG into the DNA and die, and HPRT$^-$ mutants do not incorporate 6-TG into their DNA and survive. Cells were grown for 4–6 wk with fresh DMEM + 6-TG replacing old medium every 3–4 d until visible colonies formed on the plates. Final plating for colony counting was carried out in two batches of three plates each.

### Statistical analysis

Statistical calculations were performed in GraphPad Prism with a statistical significance of $P < 0.05$. Statistical significance was determined using repeated one-way ANOVA.

## Supplementary Information

## Acknowledgements

This work was supported by Hollings Cancer Center Postdoctoral Fellowship to S Grelet; National Insitutes of Health (NIH) National Center for Advancing Translational Sciences TL1 TR001451 and UL1 TR001450 to JA Karam; and NIH National Cancer Institute CA154663 to PH Howe. We thank Prof Daniel Christ, Garvan Institute of Medical Research (Sydney), for providing the iMab scFv antibody fragment.

### Author Contributions

B Mohanty: conceptualization, formal analysis, investigation, methodology, and writing—original draft.
JA Karam: conceptualization, formal analysis, validation, investigation, methodology, and writing—review and editing.
BV Howley: conceptualization, investigation, and writing—review and editing.
A Dalton: investigation and writing—review and editing.
S Grelet: formal analysis, investigation, and methodology.
T Dincman: investigation.
WS Streitfeld: investigation.
J-H Yoon: investigation.
L Balakrishnan: resources.

WJ Chazin: resources.

DT Long: resources and writing—original draft, review, and editing.

PH Howe: conceptualization, resources, supervision, funding acquisition, and writing—review and editing.

## Conflict of Interest Statement

The authors declare that they have no conflict of interest.

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
