## [Reviewer comments · Life Science Alliance]

Life Science Alliance

Heterogeneous nuclear ribonucleoprotein E1 binds polycytosine DNA and monitors genome integrity

Philip Howe, Bidyut Mohanty, Joseph Karam, Breege Howley, Annamarie Dalton, Simon Grelet, Toros Dincman, William Streitfeld, Je-Hyun Yoon, Lata Balakrishnan, Walter Chazin, and David Long
DOI: <https://doi.org/10.26508/lsa.202000995>

Corresponding author(s): Philip Howe, Medical University of South Carolina

Review Timeline:

Submission Date:	2020-12-15
Editorial Decision:	2021-03-22
Revision Received:	2021-06-22
Editorial Decision:	2021-06-24
Revision Received:	2021-07-01
Accepted:	2021-07-02

Transaction Report:

March 22, 2021

Re: Life Science Alliance manuscript #LSA-2020-00995-T

Prof. Philip H Howe
Medical University of South Carolina
Hollings Cancer Center.
173 Ashley Ave. Room 501A
Charleston, SC 29425

Dear Dr. Howe,

Thank you for submitting your manuscript entitled "Heterogeneous nuclear ribonucleoprotein E1 binds polycytosine DNA tracts and monitors genome integrity." to Life Science Alliance. The manuscript was assessed by expert reviewers, whose comments are appended to this letter.

We apologize for this extended and unusual delay in getting back to you. As you will note from the reviewers' comments below, the reviewers are quite enthusiastic about these findings. Thus, we encourage you to submit a revised version of the manuscript to LSA that addresses all of the reviewers' points.

Thank you for this interesting contribution to Life Science Alliance. We are looking forward to receiving your revised manuscript.

Sincerely,

Shachi Bhatt, Ph.D.

Executive Editor

Life Science Alliance

<https://www.lsjournal.org/>

Interested in an editorial career? EMBO Solutions is hiring a Scientific Editor to join the international Life Science Alliance team. Find out more here -

https://www.embo.org/documents/jobs/Vacancy_Notice_Scientific_editor_LSA.pdf

B. MANUSCRIPT ORGANIZATION AND FORMATTING:

Reviewer #1 (Comments to the Authors (Required)):

In this excellent study, Mohanty et al. explored the new functions of hnRNP E1 by examining its ability to bind poly-C DNA tracts and its role in the maintenance of genome integrity. hnRNP E1 was previously known to bind poly-C RNA tracts and has multiple functions, such as maintaining mRNA stability, regulating alternative pre-mRNA splicing, and inhibiting protein translation. In this study, the authors show that hnRNP E1 is able to bind to poly-C DNA tracts in the promoters of several

oncogenes and telomeres, with equal efficiency to poly-C RNA tracts. They further show that hnRNP E1 is able to bind and stabilize secondary structures i-motifs generated by poly-C DNA tracts and indirectly regulates the formation of secondary structures G4s on the complementary DNA strand. Knockdown of hnRNP E1 leads to accumulation of G4s and a significant increase of DNA damage signals colocalizing with the G4s. The authors then show that hnRNP E1 physically interacts with replication protein A, that hnRNP E1 is able to activate the flap endonuclease Fen1, and that hnRNP E1 inhibits PCNA monoubiquitination, all of which are important for error-free DNA repair and genome integrity maintenance.

Overall, this manuscript is well written. This study revealed additional functions of the multifunctional protein hnRNP E1. These new discoveries are of major significance to the field, and therefore this paper should be of interest to the broad audience of Life Science Alliance.

One major comment:

While the authors provided convincing data to demonstrate that hnRNP E1 is able to bind to poly-C DNA tracts and that hnRNP E1 is involved in DNA damage repair, the authors did not establish a strong link between its poly-C DNA tract binding function and its DNA repair related function. It still remains unknown whether the DNA damage/repair sites are near the poly-C DNA tracts. The authors did show immunofluorescence data of colocalization of i-motifs/hnRNP E1 (Fig. 2J) and colocalization of γ -H2AX/G4s (Fig. 3D), but these are indirect evidence and in Fig. 2J a significant portion of i-motifs did not colocalize with hnRNP E1. Therefore, it would help strengthen the conclusions of this study if the authors could perform ChIP-PCR assays (using a γ -H2AX antibody) to confirm that the DNA damage/repair sites are near the poly-C DNA tracts (using the promoters of oncogenes c-MYC and h-RAS as examples).

Minor comments:

1. Fig. 3G, it seems that RPA binds to c-MYC C more efficiently than hnRNP E1 does. This result is contradictory to the illustration in Fig. 5 showing hnRNP E1 (instead of RPA) binding to the poly-C DNA tract. If RPA has a higher binding efficiency, then RPA should bind the poly-C DNA tract first.
2. On page 6, the description of the results of Fig. 1M does not match the actual data in the figure panel. For example, "a 5-6 fold excess protein over TEL DNA C" whereas the figure panel actually shows 4.4 fold. "a 12 fold excess protein over BAT RNA was needed" whereas the figure panel actually shows 11 fold. And "30-fold molar excess of the protein resulted in 30% shift of VEGF" whereas the figure panel actually shows 25% shift of VEGF.
3. Fig. 3E, a western blot of RPA is needed to confirm the increased RPA levels in E1KD.
4. On page 8, the description of the results in Fig. 2B and Table S2 needs elaboration. "Approximately 12.6% of the upregulated and 18.3% of the downregulated transcripts in E1KD cells matched with promoter-proximal sequences (Fig. 2B). All high 'C' sites identified by this analysis do not contain tracts of poly-C; sequences that contain multiple 'CCC' tracts, each with minimum 3 'C's were identified and are highlighted in Table S2." --- Are all of the 3080 high "C" sites in Fig. 2B promoter-proximal sequences? And none of the 3080 high "C" sites have poly-C tracts? If none of them have poly-C tracts, then how were the "sequences that contain multiple 'CCC' tracts" identified?
5. Fig. 2H, the labels on the top and the gel lanes are misaligned.

6. Fig. S4, there are four figure panels, but the figure legend describes only three.

Reviewer #2 (Comments to the Authors (Required)):

This manuscript by Mohanty et al showcase polyC recognition by hnRNP E1. Detailed DNA binding experiments show specific recognition of poly C tracts by the E1 protein. Cellular data complement the biochemical data and show both stimulation of DDR and a link to genomic instability. However, the connections made in the paper with respect to interactions with RPA and stimulation of FEN1 activity need more work. I recommend that the authors leave out the FEN1 and RPA data and focus on the poly C recognition data. The authors would be better served to obtain more concrete data on RPA and FEN1 and possibly communicate those activities as a separate ms when adequate data have been collected to support those findings.

1. Abstract: The following line is missing a verb: "The protein interacted physically and functionally with replication protein A and (?) activity of the facilitated flap endonuclease Fen1 at polycytosine tracts on DNA."
2. Typo in introduction: "RPA coats ssDNA to protects it and in addition, it leads to several processes as described below".
3. Reference cited twice: "(Wold, 1997; Caldwell & Spies, 2020; Sugitani & Chazin, 2015; Caldwell & Spies, 2020; Maréchal & Zou, 2015).
4. Typo: Okazaki fragment maturation (OFM) involves removal of single-stranded RNA-DNA flap by Fen1 endonuclease and RNase HI, which is (are) regulated by RPA.
5. Missing a conjunction: Hydroxyurea (HU) treatment of cells reduces nucleotide pool in the cell and can generate stretches of ssDNA because helicase and DNA polymerase can be uncoupled; thus, ssDNA binding proteins such as RPA play important role in protecting the ssDNA
6. Typo, remove capital I: All these findings underscore the importance of RPA and other ssDNA binding proteins in DNA integrity.
7. Should this read RPA foci?: "We analyzed RPA localization by immunofluorescence and observed that E1KD cells displayed increased accumulation of RPA foci (Fig. 3E).
8. Is this allowed? I think this would be a key piece of information to add to this ms as it describes binding between the two proteins: "separate mass spectrometric analysis of hnRNP E1-bound proteins revealed its interaction with RPA subunit Rpa3 (Howley and Howe, unpublished)."
9. Fig. 3G. This particular EMSA does not necessarily show physical interaction between the two proteins. Depending on the length of the oligonucleotide substrate (27 nt in the case of the c-myc substrate), both proteins can adequately occupy the same oligo, without physically interacting with each other. Please remove this inference of cooperative binding between the two proteins.
10. Fig 3L. Why is there residual cleavage in the experiment in the absence of Fen1. This basal cleavage increases in a concentration dependent manner. The authors need to address this. They might be missing a latent cleavage activity or an intrinsic contaminant that is interacting with hnRNP1. Can the authors show a SDS-PAGE of their purified proteins used in the study? A necessary control here is one of non-specificity. The stimulation of the Fen1 by hnRNP1 could also be explained by the protein structurally binding to ssDNA and passively promoting cleavage. The usual control here is adding the bacterial SSB protein and showing a lack of stimulation, or adding a version of hnRNP1 that does not bind to DNA. Maybe the authors can add the individual KH domains to this reaction and test this idea further.

The authors have shown hnRNP1 binding to C-rich substrates. This appears to be a follow of previous studies that show this activity. The conclusion that RPA and hnRNP1 work together needs to be further established.

1. Since the authors have GST-hnRNP1, they should at a minimum test if it interacts physically with RPA in the absence of DNA. Since the MS data suggest this interaction, the authors should experimentally test this interaction.
2. The physical interaction between the proteins should be perturbed and then the effect of cooperative binding with hnRNP1 on DNA should be tested.

The conclusion that hnRNP1 enhances the activity of Fen1 also needs to be further established.

1. Similar to RPA, the physical interaction between the two proteins needs to be tested.
2. Specificity in the cleavage reactions needs to be established. Adding a ssDNA binding protein to any reaction will influence a second DNA processing enzyme.

The data suggesting that hnRNP1 binds to poly-C tracts by itself is quite strong and I believe the incomplete experiments with RPA and Fen1 are not necessary for this manuscript (and they are circumstantial at best). Leaving out the RPA and Fen1 data might allow the authors to focus on polyC recognition in different contexts. For example, the authors could expand on how hnRNP1 would bind to a R-loop with a poly-C, or even the Fen1 flap substrate in a poly-C context. Obtaining good measures of the thermodynamics of the interactions (Kd's etc) would also be useful in establishing the binding properties of hnRNP1.

Reviewer #3 (Comments to the Authors (Required)):

Mohanty et.al., in their manuscript titled, "Heterogeneous nuclear ribonucleoprotein E1 binds polycytosine DNA tracts and monitors genome integrity" describe how a tumor suppressor protein, Heterogeneous nuclear ribonucleoprotein E1 (hnRNP E1) maintains DNA integrity by regulating DNA damage signaling, suppressing mutation formation, maintaining DNA secondary structures, and telomere maintenance. The hnRNP E1 is originally ascribed to bind to RNA sequences to facilitate mRNA stability, alternative splicing, and suppressing protein translation of several metastasis-associated mRNAs. In the same vein, the authors show hnRNP E1 binds to polycytosine-rich DNA tracts of promoters of several oncogenes and telomeres, thereby aiding in genome integrity. Overall, the manuscript is well written, with a cohesive background, sound rationale, and well-delineated experimentation. Most experiments are rigorous (with a few minor suggestions below to strengthen their main argument) and conclusions well-derived. The discussion section aptly describes a discourse on the observations in the paper. The authors are congratulated on deriving sound and important observations.

1. It may be prudent to show in supplement the binding data for double-stranded DNA, rather than reference it as 'data not shown'. This rationale speaks to the argument that despite the possibility of intra-strand structure formation at dinucleotide-type structures to which the hnRNP E1 can bind, there is clear affinity and possible functional implication of single-strand specific binding as seen in all the earlier observations.
2. In figure 3b and c, please correct 'gH2AX' to 'γH2AX'. Likewise, please confirm in figure 4I, the concentration of HU- is it millimolar or micromolar?
3. Most data on genomic instability are derived from work on one human cell line-A549, a lung cancer-specific model cell line. Can the authors rationalize the use of this particular cell line over other cancer cell lines?

No other issues noted.

Reviewer #1

In this excellent study, Mohanty et al. explored the new functions of hnRNP E1 by examining its ability to bind poly-C DNA tracts and its role in the maintenance of genome integrity. hnRNP E1 was previously known to bind poly-C RNA tracts and has multiple functions, such as maintaining mRNA stability, regulating alternative pre-mRNA splicing, and inhibiting protein translation. In this study, the authors show that hnRNP E1 is able to bind to poly-C DNA tracts in the promoters of several oncogenes and telomeres, with equal efficiency to poly-C RNA tracts. They further show that hnRNP E1 is able to bind and stabilize secondary structures i-motifs generated by poly-C DNA tracts and indirectly regulates the formation of secondary structures G4s on the complementary DNA strand. Knockdown of hnRNP E1 leads to accumulation of G4s and a significant increase of DNA damage signals colocalizing with the G4s. The authors then show that hnRNP E1 physically interacts with replication protein A, that hnRNP E1 is able to activate the flap endonuclease Fen1, and that hnRNP E1 inhibits PCNA monoubiquitination, all of which are important for error-free DNA repair and genome integrity maintenance.

Overall, this manuscript is well written. This study revealed additional functions of the multifunctional protein hnRNP E1. These new discoveries are of major significance to the field, and therefore this paper should be of interest to the broad audience of Life Science Alliance.

One major comment:

While the authors provided convincing data to demonstrate that hnRNP E1 is able to bind to poly-C DNA tracts and that hnRNP E1 is involved in DNA damage repair, the authors did not establish a strong link between its poly-C DNA tract binding function and its DNA repair related function. It still remains unknown whether the DNA damage/repair sites are near the poly-C DNA tracts. The authors did show immunofluorescence data of colocalization of i-motifs/hnRNP E1 (Fig. 2J) and colocalization of γ -H2AX/G4s (Fig. 3D), but these are indirect evidence and in Fig. 2J a significant portion of i-motifs did not colocalize with hnRNP E1. Therefore, it would help strengthen the conclusions of this study if the authors could perform ChIP-PCR assays (using a γ -H2AX antibody) to confirm that the DNA damage/repair sites are near the poly-C DNA tracts (using the promoters of oncogenes c-MYC and h-RAS as examples).

Response: *We have conducted ChIP-PCR using γ -H2AX antibody as suggested by the reviewer. As shown in Fig. 2 F, γ -H2AX was localized to c-MYC promoter region in E1KD cells, which was enhanced upon HU exposure. This strongly suggests hnRNP E1 protects DNA from DNA damage.*

Minor comments:

1. Fig. 3G, it seems that RPA binds to c-MYC C more efficiently than hnRNP E1 does. This result is contradictory to the illustration in Fig. 5 showing hnRNP E1 (instead of RPA) binding to the poly-C DNA tract. If RPA has a higher binding efficiency, then RPA should bind the poly-C DNA tract first.

Response: *We thank the reviewer for pointing this out, accordingly, we have modified Figure 5. At present we do not know if and how many sites are there to which hnRNP E1 binds more strongly than RPA does and vice versa. It is possible that both possibilities occur. Therefore, the figure is modified to show RPA and hnRNP E1 together bind to poly-C single-stranded DNA in a coordinated fashion. It may be noted that we have removed data on RPA and Fen1 from the manuscript as per the suggestions of Reviewer 2.*

2. On page 6, the description of the results of Fig. 1M does not match the actual data in the figure panel. For example, "a 5-6 fold excess protein over TEL DNA C" whereas the figure panel actually shows 4.4 fold. "a 12 fold excess protein over BAT RNA was needed" whereas the figure panel actually shows 11 fold. And "30-fold molar excess of the protein resulted in 30% shift of VEGF" whereas the figure panel actually shows 25% shift of VEGF.

Response: *We made all changes in the text according to data in the figures suggested by the reviewer.*

3. Fig. 3E, a western blot of RPA is needed to confirm the increased RPA levels in E1KD.

Response: *We thank the reviewer for the suggestion; however, we have removed all data associated with RPA and Fen1 as suggested by Reviewer 2.*

4. On page 8, the description of the results in Fig. 2B and Table S2 needs elaboration. "Approximately 12.6% of the upregulated and 18.3% of the downregulated transcripts in E1KD cells matched with promoter-proximal sequences (Fig. 2B). All high 'C' sites identified by this analysis do not contain tracts of poly-C; sequences that contain multiple 'CCC' tracts, each with minimum 3 'C's were identified and are highlighted in Table S2." --- Are all of the 3080 high "C" sites in Fig. 2B promoter-proximal sequences? And none of the 3080 high "C" sites have ploy-C tracts? If none of them have ploy-C tracts, then how were the "sequences that contain multiple 'CCC' tracts" identified?

Response: *Yes, all of the 3080 high "C" sites in Fig. 2B are in promoter-proximal region. After identifying high "C" sites we have manually identified sites containing multiple "CCC" sequences.*

5. Fig. 2H, the labels on the top and the gel lanes are misaligned.

Response: *The figure and the labels are now aligned.*

6. Fig. S4, there are four figure panels, but the figure legend describes only three.

Response: *The figure legends have been corrected according to the figure.*

Reviewer #2

This manuscript by Mohanty et al showcase polyC recognition by hnRNP E1. Detailed DNA binding experiments show specific recognition of poly C tracts by the E1 protein. Cellular data complement the biochemical data and show both stimulation of DDR and a link to genomic instability. However, the connections made in the paper with respect to interactions with RPA and stimulation of FEN1 activity need more work. I recommend that the authors leave out the FEN1 and RPA data and focus on the poly C recognition data. The authors would be better served to obtain more concrete data on RPA and FEN1 and possibly communicate those activities as a separate ms when adequate data have been collected to support those findings.

Response: *We thank the reviewer to give us deep insight into the physical and/or functional interactions between hnRNP E1 and RPA, and hnRNP E1 and Fen1. According to the reviewer's suggestion we have removed the data on RPA and Fen1 (please see later part of the response to the reviewer's comments also).*

1. Abstract: The following line is missing a verb: "The protein interacted physically and functionally with replication protein A and (?) activity of the facilitated flap endonuclease Fen1 at polycytosine tracts on DNA."

Response: *Because we removed data on RPA and Fen1 according to the reviewer's suggestion the above-mentioned line has been removed from the abstract.*

2. Typo in introduction: "RPA coats ssDNA to protects it and in addition, it leads to several processes as described below".

Response: *The typo "protects" was changed to "protect".*

3. Reference cited twice: "(Wold, 1997; Caldwell & Spies, 2020; Sugitani & Chazin, 2015; Caldwell & Spies, 2020; Maréchal & Zou, 2015).

Response: *The duplicate reference is deleted.*

4. Typo: Okazaki fragment maturation (OFM) involves removal of single-stranded RNA-DNA flap by Fen1 endonuclease and RNase HI, which is (are) regulated by RPA.

Response: *The typo is corrected.*

5. Missing a conjunction: Hydroxyurea (HU) treatment of cells reduces nucleotide pool in the cell and generates stretches of ssDNA due to uncoupling of helicase and DNA polymerase; thus, ssDNA binding proteins such as RPA play important role in protecting the ssDNA.

Response: *Sentence is now modified.*

6. Typo, remove capital I: All these findings underscore the Importance of RPA and other ssDNA binding proteins in DNA integrity.

Response: *The typo has been corrected.*

7. Should this read RPA foci?: "We analyzed RPA localization by immunofluorescence and observed that E1KD cells displayed increased accumulation of RPA foci (Fig. 3E).

Response: *Although the correction could have been made, we have completely removed all data on RPA as was suggested by the reviewer.*

8. Is this allowed? I think this would be a key piece of information to add to this ms as it describes binding between the two proteins: "separate mass spectrometric analysis of hnRNP E1-bound proteins revealed its interaction with RPA subunit Rpa3 (Howley and Howe, unpublished)."

Response: *We have removed all data on RPA and Fen1 as per the reviewer's suggestions; therefore, we did not address the issue.*

9. Fig. 3G. This particular EMSA does not necessarily show physical interaction between the two proteins. Depending on the length of the oligonucleotide substrate (27 nt in the case of the c-myc substrate), both proteins can adequately occupy the same oligo, without physically interacting with each other. Please remove this inference of cooperative binding between the two proteins.

Response: *As per the reviewer's suggestion we have removed data on RPA and Fen1 from the manuscript and are expanding these findings for a future manuscript. We hope that this is acceptable to the reviewer, other reviewers, and the editor.*

10. Fig 3L. Why is there residual cleavage in the experiment in the absence of Fen1. This basal cleavage increases in a concentration dependent manner. The authors need to address this. They might be missing a latent cleavage activity or an intrinsic contaminant that is interacting with hnRNP1. Can the authors show a SDS-PAGE of their purified proteins used in the study? A necessary control here is one of non-specificity. The stimulation of the Fen1 by hnRNP1 could also be explained by the protein structurally binding to ssDNA and passively promoting cleavage. The usual control here is adding the bacterial SSB protein and showing a lack of stimulation, or adding a version of hnRNP1 that does not bind to DNA. Maybe the authors can add the individual KH domains to this reaction and test this idea further.

Response: *As per the reviewer's suggestion we have removed data on Fen1 (as well as on RPA) from the manuscript and are expanding these findings for a future manuscript. We hope that this is acceptable to the reviewer, other reviewers, and the editor.*

The authors have shown hnRNP1 binding to C-rich substrates. This appears to be a follow of previous studies that show this activity. The conclusion that RPA and hnRNP1 work together needs to be further established.

1. Since the authors have GST-hnRNP1, they should at a minimum test if it interacts physically with RPA in the absence of DNA. Since the MS data suggest this interaction, the authors should experimentally test this interaction.

2. The physical interaction between the proteins should be perturbed and then the effect of cooperative binding with hnRNP1 on DNA should be tested.

The conclusion that hnRNP1 enhances the activity of Fen1 also needs to be further established.

1. Similar to RPA, the physical interaction between the two proteins needs to be tested.

2. Specificity in the cleavage reactions needs to be established. Adding a ssDNA binding protein to any reaction will influence a second DNA processing enzyme.

The data suggesting that hnRNP1 binds to poly-C tracts by itself is quite strong and I believe the incomplete experiments with RPA and Fen1 are not necessary for this manuscript (and they are circumstantial at best). Leaving out the RPA and Fen1 data might allow the authors to focus on polyC recognition in different contexts. For example, the authors could expand on how hnRNP1 would bind to a R-loop with a poly-C, or even the Fen1 flap substrate in a poly-C context. Obtaining good measures of the thermodynamics of the interactions (Kd's etc) would also be useful in establishing the binding properties of hnRNP1.

Response: *We thank the reviewer to give us deep insight into the physical and/or functional interactions between hnRNP E1 and RPA, and hnRNP E1 and Fen1. According to the reviewer's suggestion we have removed all data on RPA and Fen1 from the manuscript. We strongly hope that this is acceptable to the reviewer, other reviewers, and the editor. We have added a new section to the figure as per the suggestions by Reviewer 1.*

Reviewer #3

Mohanty et.al., in their manuscript titled, "Heterogeneous nuclear ribonucleoprotein E1 binds polycytosine DNA tracts and monitors genome integrity" describe how a tumor suppressor protein, Heterogeneous nuclear ribonucleoprotein E1 (hnRNP E1) maintains DNA integrity by regulating DNA damage signaling, suppressing mutation formation, maintaining DNA secondary structures, and telomere maintenance. The hnRNP E1 is originally ascribed to bind to RNA sequences to facilitate mRNA stability, alternative splicing, and suppressing protein translation of several metastasis-associated mRNAs. In the same vein, the authors show hnRNP E1 binds to polycytosine-rich DNA tracts of promoters of several oncogenes and telomeres, thereby aiding in genome integrity. Overall, the manuscript is well written, with a cohesive background, sound rationale, and well-delineated experimentation. Most experiments are rigorous (with a few minor suggestions below to strengthen their main argument) and conclusions well-derived. The discussion section aptly describes a discourse on the observations in the paper. The authors are congratulated on deriving sound and important observations.

Response: *We thank the reviewer for the nice comments. Please find responses to all suggestions below.*

1. It may be prudent to show in supplement the binding data for double-stranded DNA, rather than reference it as 'data not shown'. This rationale speaks to the argument that despite the possibility of intra-strand structure formation at dinucleotide-type structures to which the hnRNP E1 can bind, there is clear affinity and possible functional implication of single-strand specific binding as seen in all the earlier observations.

Response: *We thank the reviewer for the suggestion. We have included the data on a double-stranded DNA of poly-C tracts is included in the supplement.*

2. In figure 3b and c, please correct 'gH2AX' to ' γ -H2AX'. Likewise, please confirm in figure 4I, the concentration of HU- is it millimolar or micromolar?

Response: *The gamma-H2AX is corrected to (gamma) γ -H2AX. HU concentration in Figure 4H is millimolar (mM).*

3. Most data on genomic instability are derived from work on one human cell line-A549, a lung cancer-specific model cell line. Can the authors rationalize the use of this particular cell line over other cancer cell lines?

Response: *Previously we had shown in a mouse cell line (NMuMG) that TGF β treatment phosphorylates and abolishes BAT RNA binding by hnRNP E1. This observation was similar to the absence of hnRNP E1 in NMuMG. This in turn causes overexpression of several metastasis genes including ILEI and induces epithelial-to-mesenchymal (EMT) transition. Recently works from other laboratories have shown similar phenomena in A549, a lung cancer-specific model, which also undergoes TGF β treatment induced EMT, and shows TGF β response with hnRNP E1 phosphorylation, ILEI overexpression through release of hnRNP E1 from its transcript, and translational activation (Song et al., 2014. Tumor Biology, 35:1377–1382; Xue et al., Tumor Biology. 2014 volume 35:7853–7859). We have observed phenotype changes in A549 cells upon hnRNP E1 knockdown (although data not shown in the manuscript, we can include a supplemental figure if suggested by the reviewer). Therefore, we conducted experiments with A549; however, we have also provided data on experiments with the mouse cell line NMuMG (Fig.2A, 2B, 2I and 4C). These data suggest a universal role for hnRNP E1 on DNA.*

No other issues noted.

June 24, 2021

RE: Life Science Alliance Manuscript #LSA-2020-00995-TR

Prof. Philip H Howe
Medical University of South Carolina
Hollings Cancer Center.
173 Ashley Ave. Room 501A
Charleston, SC 29425

Dear Dr. Howe,

Thank you for submitting your revised manuscript entitled "Heterogeneous nuclear ribonucleoprotein E1 binds polycytosine DNA and monitors genome integrity". We would be happy to publish your paper in Life Science Alliance pending final revisions necessary to meet our formatting guidelines.

- please upload your main and supplementary figures as single files
- please note that titles in the system and manuscript file must match
- please be sure that all Authors inserted in the Authors Contribution section
- please add supplementary figures and table legends to the main manuscript text after the main figure legends;
- please add callouts for Figures S1A-H, S2A, B, and S4D to your main manuscript text;
- there is a callout for figure S2C although the actual figure and its legend don't have this panel. Please correct
- we encourage you to revise the figure legend for figure S4 such that the figure panels are introduced in alphabetical order

Figure checks:

- figure 6 is wrongly labeled (it should be 5)
- please indicate molecular weights next to each protein blot
- the scale bars for Figures 1B; 2I, J, K; 3A and D; S4D are hardly readable. Please improve readability or indicate the size in the figure legend.

A. FINAL FILES:

B. MANUSCRIPT ORGANIZATION AND FORMATTING:

Sincerely,

July 2, 2021

RE: Life Science Alliance Manuscript #LSA-2020-00995-TRR

Prof. Philip H Howe
Medical University of South Carolina
Hollings Cancer Center.
173 Ashley Ave. Room 501A
Charleston, SC 29425

Dear Dr. Howe,

Thank you for submitting your Research Article entitled "Heterogeneous nuclear ribonucleoprotein E1 binds polycytosine DNA and monitors genome integrity". It is a pleasure to let you know that your manuscript is now accepted for publication in Life Science Alliance. Congratulations on this interesting work.

DISTRIBUTION OF MATERIALS:

Again, congratulations on a very nice paper. I hope you found the review process to be constructive and are pleased with how the manuscript was handled editorially. We look forward to future exciting submissions from your lab.

Sincerely,
